# Human Respiratory Syncytial Virus Infection in a Human T Cell Line Is Hampered at Multiple Steps

**DOI:** 10.3390/v13020231

**Published:** 2021-02-02

**Authors:** Ricardo de Souza Cardoso, Rosa Maria Mendes Viana, Brenda Cristina Vitti, Ana Carolina Lunardello Coelho, Bruna Laís Santos de Jesus, Juliano de Paula Souza, Marjorie Cornejo Pontelli, Tomoyuki Murakami, Armando Morais Ventura, Akira Ono, Eurico Arruda

**Affiliations:** 1Department of Cell and Molecular Biology, School of Medicine of Ribeirao Preto, University of Sao Paulo, São Paulo 14049-900, Brazil; rdsc@umich.edu (R.d.S.C.); rosamendes1993@gmail.com (R.M.M.V.); brendavitti@usp.br (B.C.V.); ana.lunardello@gmail.com (A.C.L.C.); brunalais@usp.br (B.L.S.d.J.); jpsouza@usp.br (J.d.P.S.); marjpontelli@usp.br (M.C.P.); 2Department of Microbiology and Immunology, University of Michigan Medical School, Ann Arbor, MI 48109, USA; tmurakam@umich.edu (T.M.); akiraono@umich.edu (A.O.); 3Department of Microbiology, Institute of Biomedical Sciences, University of Sao Paulo, Sao Paulo 05508-000, Brazil; amventur@icb.usp.br

**Keywords:** human respiratory syncytial virus (HRSV) infection in T-cell line, HRSV R18 fusion assay, HRSV intracellular trafficking, HRSV inclusion body-associated granules (IBAG′s), inefficient HRSV replication A3.01, HRSV filament formation, HRSV low protein production, HRSV entry

## Abstract

Human respiratory syncytial virus (HRSV) is the most frequent cause of severe respiratory disease in children. The main targets of HRSV infection are epithelial cells of the respiratory tract, and the great majority of the studies regarding HRSV infection are done in respiratory cells. Recently, the interest on respiratory virus infection of lymphoid cells has been growing, but details of the interaction of HRSV with lymphoid cells remain unknown. Therefore, this study was done to assess the relationship of HRSV with A3.01 cells, a human CD4^+^ T cell line. Using flow cytometry and fluorescent focus assay, we found that A3.01 cells are susceptible but virtually not permissive to HRSV infection. Dequenching experiments revealed that the fusion process of HRSV in A3.01 cells was nearly abolished in comparison to HEp-2 cells, an epithelial cell lineage. Quantification of viral RNA by RT-qPCR showed that the replication of HRSV in A3.01 cells was considerably reduced. Western blot and quantitative flow cytometry analyses demonstrated that the production of HRSV proteins in A3.01 was significantly lower than in HEp-2 cells. Additionally, using fluorescence in situ hybridization, we found that the inclusion body-associated granules (IBAGs) were almost absent in HRSV inclusion bodies in A3.01 cells. We also assessed the intracellular trafficking of HRSV proteins and found that HRSV proteins colocalized partially with the secretory pathway in A3.01 cells, but these HRSV proteins and viral filaments were present only scarcely at the plasma membrane. HRSV infection of A3.01 CD4^+^ T cells is virtually unproductive as compared to HEp-2 cells, as a result of defects at several steps of the viral cycle: Fusion, genome replication, formation of inclusion bodies, recruitment of cellular proteins, virus assembly, and budding.

## 1. Introduction

Human respiratory syncytial virus (HRSV), of the family *Pneumoviridae*, is a common respiratory pathogen that circulates worldwide. HRSV is a major cause of serious lower respiratory tract disease, mainly bronchiolitis in children, and of severe disease in the elderly [1]. HRSV infects mainly epithelial cells of the respiratory tract [1] but has also been detected in nonrespiratory tissues and cells [2,3]. In that regard, HRSV and other respiratory viruses have been detected in tonsillar tissues and respiratory secretions from children with tonsillar hypertrophy without symptoms of acute respiratory infection [4], suggesting that HRSV may infect secondary lymphoid tissues. In addition, HRSV antigen has been detected in circulating T lymphocytes. Thus, it is conceivable that it may affect immune function [5].

The HRSV genome is a single-stranded RNA with 10 genes that encode 11 proteins [1]. Virus entry is mediated by the G protein that binds to the host cell [6], resulting in the fusion between the viral envelope and cell membrane performed by the HRSV F protein [7,8]. It is also known that HRSV enters cells by macropinocytosis, a process in which nucleolin participates as virus receptor [9,10]. HRSV replication involves the production of inclusion bodies (IBs) [11], where viral proteins L, P, N, M, and M1,2 are present [12,13,14]. However, the subsequent HRSV assembly process is not entirely understood. It has been shown that the viral glycoproteins follow the secretory pathway [15,16,17,18] to reach the plasma membrane but also utilize Apical Recycling Endosome (ARE) machinery during this process [19]. In contrast, the trafficking of HRSV non-glycosylated proteins during virus assembly remains quite unclear. It is known that HRSV M protein forms dimers prior to participating in the formation of viral progeny [20], and that this protein has affinity to endomembrane system [21]. HRSV N protein coats the virus genome and forms inclusion bodies, an essential role in which it is helped by the P protein [22]. The inclusion bodies of the HRSV are places where the viral replication and transcription occur. In addition, the IBs are responsible for stabilizing the viral mRNAs to confer an efficient protein translation process [14]. HRSV N protein was also observed to colocalize partially with the Golgi [23]. Consistent with these findings, we recently showed that, in HEp-2 cells, the M and N engage partially with secretory pathway and with the retromer complex [24]. Furthermore, HRSV P protein was found partially colocalizing with endosomal vesicles [25]. Together, these findings suggest the possibility that the non-glycosylated proteins of the HRSV reach the assembly sites at least partially using secretory and/or endosomal pathways. The steps of viral assembly and budding take place at the plasma membrane of the infected cells [21,26,27]. It is known that the HRSV F protein cytoplasmic tail is pivotal in the virus budding process [7,28]. Finally, the HRSV budding process results in a filamentous viral particle, an event dependent on Rab11-FIP2 protein but not Vps4 [29].

Knowledge on HRSV-cell interactions has accumulated through studies that have used models based on epithelial cell cultures susceptible and permissive to HRSV infection. Nonetheless, virus-host cell interactions may be different between cells of epithelial origin such as HEp-2, in which virus progeny production results in cell death, and lymphomononuclear cells, in which HRSV may cause long-term or persistent infection. Previous studies have shown that HRSV infects CD4^+^ T cells [5] and Breg cells of neonates [30], which may enhance HRSV disease. Very little is known about HRSV replication and progeny production in lymphoid cells, hence the present study was done to investigate intracellular HRSV assembly and replication in the human CD4^+^ T cell line A3.01, which belong to the lymphoid cell lineage.

## 2. Materials and Methods

### 2.1. Cells and Viruses

The human CD4^+^ T cell line A3.01 was obtained from the AIDS Research Reagent Program and maintained in Rosswell Park Memorial Institute (RPMI) culture medium, supplemented with 10% of fetal bovine serum (FBS) and 1% of antimycotic/antibiotic. HEp-2 and Vero cells were acquired from ATCC and maintained in MEM with 10% of FBS in 1% of antimycotic/antibiotic. The virus used for this study was HRSV A long strain (ATCC), propagated in HEp-2 cells and titrated in Vero cells, following routine agar-based plaque assays. The multiplicity of infection (MOI) used for the infections are specified in the experiment results.

### 2.2. Antibodies

The antibodies used for immunofluorescence were FITC-conjugated mouse monoclonal anti-RSV N (MAB 8583F, Millipore, Burlington, MA, USA) diluted 1:60. Mouse monoclonal anti-RSV F (MAB8262X, Millipore) was diluted 1:100, mouse anti-CD63 (Clone H5C6 RUO, BD Pharmingen, Franklin Lakes, NJ, USA) was diluted 1:100, and mouse polyclonal anti-RSV M [12] was diluted 1:50. Rabbit polyclonal antibodies anti-TGN46 (ABT95, Millipore, Burlington, MA, USA), anti-Giantin (PRB-114C-200, Covance, Princeton, NJ, USA), anti-SNX2 [31], and anti-Lamp-1 D2D11 XPR (9091T, Cell Signaling) were diluted, respectively, 1:300, 1:500, 1:200, and 1:750. Mouse monoclonal anti-EEA1 (610457-BD) was diluted 1:200. Alexa Fluor 594-conjugated goat anti-rabbit (ab150080, Abcam, Cambridge, UK) was diluted 1:500. Alexa Fluor 647-conjugated goat anti-mouse (ab150115, Abcam) was diluted 1:2000. The antibodies for western blot experiments were goat anti-RSV (Abcam ab20745), diluted 1:1000; HRP-conjugated goat anti-rabbit (Invitrogen, Carlsbad, CA, USA), diluted 1:5000; and HRP-conjugated rabbit anti-goat (305 035 003, Jackson ImmunoResearch, West Grove, PA, USA), and rabbit polyclonal anti-GAPDH (G9595, Sigma, St. Louis, MO, USA), diluted 1:5000. All the antibodies for immunofluorescence were diluted in phosphate-buffered saline 1X (PBS 1X) with 1% bovine serum albumin (BSA) (PBS-BSA). The antibodies for western blotting were diluted in PBS 1X containing 0,1% of Tween 20 and 0,01% of sodium azide.

### 2.3. Flow Cytometry

HEp-2 or A3.01 cells were infected with HRSV, and the harvest occurred 24 h, 48 h, and 72 h post-infection, when cells were fixed with 4% paraformaldehyde (PFA) for 20 min. After that, cells were permeabilized in a solution of 0.2% Triton-100X for 5 min, and incubated with FITC-labeled mouse anti-RSV Nucleoprotein (Millipore, Burlington, MA, USA), diluted 1:100 in PBS-BSA, followed by 3 washes with PBS-BSA. The cells were analyzed in a BD LSR Fortessa flow cytometer, and the results were analyzed in the Flow Jo software. The 2D plot, in which the plans were fluorescence signals for N versus forward scatter, was used as a gating strategy to perform the single-cell analysis for N-positive population to establish the mean fluorescence intensity. Experiments were done 3 or more times independently.

### 2.4. Biobond and Poly-Lysine Coverslips Treatment

Coverslips were treated with 0.1 N HCl in 100% ethanol. Then, they were treated in 100% alcohol, air-dried, and treated with 4% Biobond (Koch Electron microscopy) in acetone for 4 min, followed by a wash in distilled water and air-drying. The same procedure was used for treatment with Poly-Lysine, but 20 µL of poly-lysine was used for incubating coverslips for 1 h and then coverslips were air-dried. T cells were deposited onto pretreated coverslips and incubated stationary for 3 h, followed by fixation in 4% PFA for 20 min, washing in PBS, and testing by immunofluorescence or fluorescent in situ hybridization (FISH).

### 2.5. Immunofluorescence

After fixation, A3.01 or HEp-2 cells on coverslips were permeabilized in 0.01% Triton X-100 in PBS 1X for 15 min and washed 5 times in PBS 1X. The cell preparations were incubated with the appropriate primary antibody for 1 h in a humidified chamber at 37 °C. After that, preparations were washed 5 times in PBS 1X, incubated with the appropriate secondary antibody and DAPI (Sigma, for 1 h), and finally washed in PBS 1X. The coverslips were mounted on slides with Flouromount and analyzed by confocal microscopy in a Leica SP5 or a Zeiss 780 microscope. To perform analysis of the quantity and size of the inclusion bodies in A3.01 and HEp-2 cells, the immunofluorescence images were subjected to Image J software analysis. Using the tool Analyze > Analyze Particles, it was possible to determine the area and number of the N-positive structures. The same procedure was done for measuring the size of the A3.01 cells, but in this case, after the analysis of the cell area with Image J software, the mean area of the A3.01 cells was calculated.

### 2.6. Fluorescent In Situ Hybridization

HEp-2 or A3.01 cells were seeded on coverslips treated or not treated with Poly-Lysine. Then, 24 h and 48 h post-infection, cells were fixed with 4% PFA and tested by FISH as per published protocol [14], with a slightly modified process. Briefly, fixed cells were permeabilized with 0.1% Triton X-100 in PBS-BSA, and then treated with free streptavidin for 1 h, refixed in 4% PFA, washed in PBS, and incubated with the hybridization mix (1 µM biotinylated poly T probe, 35% formamide, 5% dextran sulfate, herring sperm DNA in 2 X SSC2) at 37 °C for 3 h, followed by serial washes in 2X SSC, SSC, and PBS 1X. After that, coverslips were incubated with Alexa Fluor 594 streptavidin, diluted 1:50 in PBS-BSA. Then, coverslips were incubated with FITC-conjugated mouse anti-RSV N (Millipore) for 1 h, diluted 1:60 in PBS-BSA. Finally, the cells were washed in PBS 1X and mounted with a mounting solution containing DAPI prior to observation using a fluorescent microscope. For counting the Inclusion bodies containing IBAGs and the IBAGs within inclusion bodies in A3.01 and HEp-2 cells, the immunofluorescence images from 3 independent experiments were subjected to manual counting. All the clear distinct granular structures within the inclusion bodies were considered IBAGs.

### 2.7. Quantitative Real-Time RT-PCR (RT-qPCR)

HEp-2 and A3.01 cells inoculated with HRSV for 1 h at 4 °C were centrifuged at 200× *g*, and part of the cells and supernatant were placed in Trizol (Invitrogen, Carlsbad, CA, USA) in a proportion of 750 µL of Trizol to 250 µL of sample for total RNA extraction according to the manufacturer’s protocol. This was considered as the timepoint zero. The remaining cells were incubated in CO_2_ at 37 °C, and aliquots of cells and supernatant were collected in Trizol (Invitrogen, Carlsbad, CA, USA) at 1 h, 3 h, 6 h, 12 h, 24 h, and 48 h post-infection. After total RNA isolation, the cDNAs were obtained by denaturation of 1 µL of total RNA for 5 min at 95 °C in the presence of 20 mM primers directed to sequences within the leader-NS1 gene of the HRSV genome. After that, tubes were placed on ice for 5 min while the reactions were prepared in a final volume of 20 µL with 2 µL of 10 × RT buffer, 0.8 µL of 100 mM dNTP, and 1 µL of MultiScribe reverse transcriptase. The reaction mixtures were incubated for 10 min at 25 °C, for 120 min at 37 °C, and for 5 min at 85 °C. The product was amplified by SYBR green RT-qPCR using specific primers (forward ACA ACA AAC TTG CGT AAA CCA AAA, reverse CCA TGC TAC TTC ATC ATT GTC AAA CA) for the HRSV leader region and (forward GCTCTTAGCAAAGTCAAGTTGAATGA, reverse TGCTCCGTTGCATGGTGTATT) for the HRSV N gene. RT-qPCR assay was done in a final volume of 10 µL with 1 µL cDNA, 20 nM forward and reverse primers, and 5 µL of SYBR green PCR master mix (Kappa, USA), with the following parameters: 50 °C for 2 min and 95 °C for 10 min, followed by 45 cycles of 95 °C for 30 s and 60 °C for 1 min. Subsequently, 1 cycle was done of 95 °C for 15 s, 60 °C for 30 s, and 95 °C for 15 s, and specific amplification was confirmed by analyzing the melting curve. All RT-qPCR assays were done on Thermocycler 7300 (Applied Biosystems, Foster City, CA, USA) and samples were set up in triplicate. The data in Figure 1B were generated using primers specific for the HRSV leader region, while the data in Figure 2 were generated using primers specific for HRSV protein N.

### 2.8. R18 Fluorescence Conjugation and Dequenching Assay

These protocols were the same as those used by Covés-Datson et al. [32] with slight modifications. HRSV or culture supernants of HEp-2 cells were incubated with 30 µg/mL of octadecyl rhodamine B chloride (R18) for 1 h at room temperature and protected from light. Then, fluorescently labeled R18-HRSV or culture supernatants of HEp-2 were separated from excess R18 by a separation column (PD-10 Desalting Columns GE Healthcare). HRSV-R18 or culture supernatants of HEp-2 were incubated in suspension with A3.01 and HEp-2 cells at 4 °C for 1 h to allow virus attachment. After that, the cells were washed 3 times with cold PBS 1X, and then incubated in a CO_2_ at 37 °C. Dequenching of R18 was measured in a Synergy Photometer at the appropriate timepoints. As positive control, 1% Triton X-100 in PBS 1X was used for total dequenching. The amount of fluorescence emitted from R18 was measured at 560 nm excitation and 590 nm emission.

### 2.9. Fluorescent Focus Assay

A3.01 cells were infected with HRSV. At 1 h, 6 h, 12 h, 24 h, and 48 h post-infection, their supernatants were collected to investigate the progeny production of the HRSV in these cells. Serial 10-fold dilutions of the supernatants collected at each timepoint were made. Then, a HEp-2 cell monolayer was infected with each of the HRSV dilutions. Three days after the infection, the cells were fixed in 4% of PFA, washed 3 times in PBS 1X, and permeabilized with 0.01% Triton X-100. After that, cells were incubated with an antibody made in mouse anti-HRSV F protein conjugated with Alexa Fluor 488, dilution 1:100. The fluorescent foci were analyzed in a fluorescent microscope and the number of foci were counted and plotted in a graph.

### 2.10. Immunoblotting

HEp-2 and A3.01 cells infected with HRSV were harvested at appropriate times post-infection using EDTA 0.05%, centrifuged at 200× *g*, suspended in lysis buffer (50 mM Tris pH 7.5, 150 mM NaCl, 10% glycerol, 5 mM EDTA, 1% Triton X-100) for 30 min on ice, and then centrifuged at × *g* for 20 min at 4 °C to obtain the cell lysate supernatant. Total protein levels were equalized using the Bio-Rad protein assay.Ssamples were mixed with sample buffer (4% SDS, 160 mM Tris-HCl pH 6.8, 20% glycerol, 100 mM DTT, and 0.1% bromophenol blue), then heated at 95 °C for 5 min. Proteins were resolved by SDS-PAGE, and then transferred onto a nitrocellulose membrane that was blocked with PBS containing 0.5% Tween 20 and 5% skim milk for 2 h. Subsequently, the nitrocellulose membrane was incubated with the primary antibody overnight at 4 °C and then incubated with secondary antibody for 1 h. Protein bands were visualized by enhanced chemiluminescence (ECL, St. Paul, MN, USA) solutions 1 (1 M Tris-HCl pH 8.5, 250 mM luminol, 90 mM *p*-coumaric acid) and 2 (30% H_2_O_2_, 1 M Tris-HCl pH 8.5) [33] and analyzed on a ChemiDoc Imaging System with the ImageLab software (Bio-Rad Laboratories, Hercules, CA, USA).

### 2.11. Statistical Analysis

Data were analyzed and plotted using GraphPad Prism 5.0 software and shown as mean ± SEM of at least 3 independent experiments. Statistical significance was determined by ANOVA or student t-test, and the *p* values were represented as * *p* < 0.05, ** *p* < 0.001, *** *p* < 0.001, and ns, not significant. Differences were considered as statistically significant if the *p* value was <0.05.

## 3. Results

### 3.1. A3.01 Lymphocytes Inoculated with HRSV Are Inefficient in Progeny Production

The A3.01 human lymphocyte cell line was infected in suspension with HRSV (MOI = 1) and analyzed by flow cytometry at several times after infection (Figure 1A). Infection was reproducibly detected in three or more independent experiments at all times post-infection, with peak at 48 h post-infection (hpi), when over 40% of the cells were infected. The highest numbers of cells positive for HRSV N protein were found at 48 hpi (Figure 1A). These results indicate that, at least under the condition used in these experiments, A3.01 cells are susceptible to HRSV infection and produce viral protein N. A3.01 cells were permissive for HRSV replication. However, virus replication in this cell type was markedly reduced and delayed in comparison with HEp-2 cells, as indicated by quantification of the HRSV genome released into the culture supernatants (Figure 1B). The fluorescent focus assay with mouse anti-HRSV F antibody indicates that infectious HRSV progeny production in A3.01 is also inefficient, with a replicative burst of less than one log_10_ from 6 to 48 h post-infection (Figure 1C).

### 3.2. HRSV Genome Replication in A3.01 Cells Is Inefficient

Since we observed that A3.01 cells were inefficient in progeny production of HRSV, we sought to investigate which step of the viral replicative cycle was compromised in these cells. We set out to assess the virus genome production by RT-qPCR targeting the HRSV N gene in HRSV-infected A3.01 and HEp-2 cells, the latter of which was cultured either as cells attached to plates (Att) on in suspension (Sus). Cells and viruses were incubated for viral adsorption for 1 h at 4 °C, the inoculum was washed away, and the cells were further incubated at 37 °C. At several times post-infection, cells were collected and processed for RNA extraction in Trizol. The same amount of virus inoculum was placed in virus-only wells, without cells, for quantification of the remaining virus inoculum after the incubation periods. The results revealed that the attachment of HRSV to HEp-2 and A3.01 cells was only slightly different. However, while the virus replicated about 10,000-fold in HEp-2 cells, we observed no replication in A3.01 cells (Figure 2). Furthermore, the amount of cell-associated HRSV RNA decreased in A3.01 cells over 3 h after inoculation, suggesting a failure in a step after virus-cell attachment such as virus-cell fusion, which would result in the degradation of internalized virions.

### 3.3. The Fusion Process of HRSV in A3.01 Cells Is Defective

The numbers of HRSV-positive A3.01 cells by flow cytometry were dependent on the duration of the incubation period for viral adsorption to the cells (Figure 3A). The fraction of cells positive for HRSV N protein at 24 hpi and 48 hpi was higher when the virus inoculum was not removed than when it was washed away after 3 h. This is consistent with the possibility that the entry of HRSV in A3.01 cells is not efficient. To test this possibility, both A3.01 and HEp-2 cell suspensions were inoculated with R18-labelled HRSV (R18-HRSV) or mixed with culture supernatants of mock-infected cells containing an equivalent amount of R18, and the virus-cell fusion was evaluated based on dequenching of R18, which occurs when viral envelope fuses with cellular membranes. We examined virus-cell fusion with 4 × 10^4^ cells per well. The virus or mock inocula were incubated with cells on ice for 1 h to allow adsorption but not entry, after which the unbound viruses were washed away. Subsequently, the virus-cell suspensions were placed at 37 °C, and the fluorescent emission by the R18 dequenching process was measured. The results revealed that, while the quantity of dequenched R18 steadily increased over time in HEp-2 cells, it remained at background level in A3.01 cells (Figure 3B). To test the possibility that the difference in the dequenching of R18 between the two cell types was due to the difference in the amounts of attached virus, we compared the attachment of R18-HRSV to HEp-2 and A3.01 cells using the same approach. Both A3.01 and HEp-2 cells were treated with 1% Triton X-100 immediately after the washes with cold PBS, thereby completely de-quenching the attached viruses (Figure 3C). The results suggested that the amounts of HRSV attached in the A3.01 and HEp-2 cells were similar (Figure 3C). Altogether, we conclude that the process of HRSV fusion to A3.01 cells was nearly abolished in comparison to HEp-2 cells.

### 3.4. HRSV Protein Production in A3.01 Cells Is Small

Even though the inefficient fusion process of the HRSV in A3.01 cells by itself could explain the dramatic differences between genome replication in A3.01 and HEp-2, we hypothesized that other steps of the HRSV production could also be affected. Since the replication of HRSV in A3.01 cells was inefficient, we examined whether the production of N protein, a main component of virus factories, was also compromised in A3.01 cells. Mean fluorescence intensity analysis of infected HEp-2 and A3.01 cells at different timepoints using flow cytometry (Figure 4A) showed that the N protein level in each HRSV-infected A3.01 cell remained close to the background level and at least 10- to a 100-fold lower than that in each HEp-2-infected cell at all times. We found the greatest difference at 48 hpi, when the quantity of protein in HEp-2 cells was approximately 42-times greater than that found in A3.01 cells (Figure 4B). Western blot analysis of HRSV proteins produced in A3.01 or HEp-2 cells revealed that only the N protein could be reliably detected in infected A3.01 cells at different times post-infection, while only faint bands revealed small amounts of the other viral proteins (Figure 4C). Although the western blotting analysis was not controlled for the number of infected cells, it revealed that the fold differences in M and P levels between HEp-2 and A3.01 were even greater than the difference in N. Therefore, not only N, which was examined in the flow cytometry, but also other viral proteins were about 10- to a 100-times less abundant in A3.01 than in HEp-2 at 48 hpi and 72 hpi. In addition, it was not even possible to see the bands corresponding to the HRSV G and F glycoproteins. These data indicate that the HRSV protein levels at different times post-infection in A3.01 are broadly compromised.

### 3.5. HRSV Inclusion Body Formation is Compromised in A3.01

Inclusion bodies (IBs) are important platforms for HRSV replication and assembly [5,16,17]. Considering that the HRSV N, P, and M are pivotal components of the IB formation and that the levels of these proteins are very low in A3.01 cells, we expected that the capacity of the HRSV to produce IBs in A3.01 cells is highly diminished. Interestingly, immunofluorescence for HRSV N protein revealed that HRSV infection induced inclusion body formation in A3.01 cells (Figure 5B,C). However, by measuring the sum of the IBs areas in A3.01, we found that they were much smaller than those in HEp-2 cells (Figure 5G, and compare Figure 5B,C to Figure 5E,F) and about 10-fold less abundant in A3.01 than in HEp-2 cells (Figure 5H). Since the area of the HEp-2 cell is approximately 3.2-times larger than A3.01 cells, the difference in the area sizes can partially account for the differences found in the quantity of HRSV N inclusion bodies in HEp-2 and A3.01. However, even when we normalized the quantity of HRSV N inclusions by the area, the quantity of HRSV N-positive IBs was higher in HEp-2 than A3.01 cells (Figure 5I).

### 3.6. The Inclusion Bodies of HRSV in A3.01 Cells Lack IBAGs

Recently, Rincheval et al. [14] showed that HRSV inclusion bodies are compartmentalized and contain ultrastructural granules, called inclusion body-associated granules (IBAGs). The authors also proposed that these structures are important for the IBs functionality, since this organization promotes stabilization of viral transcripts and hence an efficient protein production process. We performed a FISH analysis to examine the formation of IBAGs of HRSV-infected A3.01 cells. A biotinylated poly-T probe was used to reveal regions enriched for polyadenylated RNA. HRSV produced IBAGs at both 24 hpi and 48 hpi in HEp-2 cells (Figure 6I–P), but in A3.01 cells, the majority of the HRSV-induced IBs did not appear to contain prominent IBAGs and therefore can be considered hypofunctional (Figure 6Q–X, pointed out by arrowheads). HRSV IBs in A3.01 cells contained staining for polyA, but such polyA signal was rarely found as clear distinct granules within the inclusion bodies as observed in HEp-2 cells. A Z-stack and quantitative analysis was performed between labelled HEp-2 (Appendix A) and A3.01 cells (Appendix A). The inclusion bodies of A3.01 and HEp-2 cells containing distinct IBAGs within were counted (Appendix A). Whereas we observed, on average, 0 or 1 infected A3.01 cells containing IBs per microscope field, the average was 15 IBAG-containing cells per field for HEp-2 cells (Appendix A). Further, while, on average, each IB contained three to four IBAGs in HEp-2 cells, no authentic IBAG could be clearly identified in the rare IBs of the infected A3.01 cells (Appendix A). Therefore, we conclude that even when poly A-containing mRNAs are present in the IBs of HRSV-infected A3.01 cells, the sequestration of such mRNAs to subcompartments of IBs is inefficient in this cell type.

### 3.7. HRSV F Protein Partially Colocalizes with Golgi Markers Giantin and TGN46 in A3.01 Cells

In epithelial cells, F protein, a transmembrane protein follows the conventional anterograde pathway from ER through Golgi to the plasma membrane [34]. Also, the Golgi participates in the intracellular traffic of the HRSV N protein to the plasma membrane [23,24]. To examine whether the trafficking of HRSV F and N proteins in A3.01 lymphocytes is similar to the one in HEp-2 cells, immunofluorescence staining was done for a cis- and medial-Golgi marker, giantin; a trans-Golgi network marker, TGN46; and the viral proteins F and N. In HRSV-infected HEp-2 cells, the viral protein F clearly colocalized with giantin and partially with TGN46 (Figure 7M–R), while in A3.01 cells, we observed no obvious association of F with these Golgi markers at 48 hpi (Figure 7E–H). Even though HRSV N protein is cytoplasmic and not transmembrane, partial colocalization with TGN46 and giantin were observed in HEp-2 cells (Figure 7S–X). However, in A3.01 cells, the N protein did not colocalize with the Golgi markers (Figure 7I–L). This suggests that while some of the HRSV F protein appears near the Golgi in A3.01 cells, there was no evident colocalization with the Golgi markers. Similar observations were made for HRSV N protein and Golgi markers. That scenario is different from HEp-2 cells, in which HRSV F protein clearly colocalized with Golgi markers, mainly giantin, and HRSV N protein, even though not clearly colocalizing with the Golgi, appeared tightly associated with Golgi markers.

### 3.8. The HRSV F and N Proteins Partially Co-Localize with Markers of Endosomal Pathway

After trafficking through the Golgi stacks, the transmembrane proteins often associate with endosomal pathways [35,36,37]. In addition, it is already known that some of the HRSV proteins follow the endosomal system to reach the plasma membrane [19,23,24,34]. To investigate whether the HRSV proteins also associate with endosomal machineries in A3.01 cells, immunofluorescence assays were performed for several markers of this pathway. First, we aimed to investigate whether HRSV proteins were preferentially targeted to late/lysosomal pathway in these cells. We assessed HRSV proteins colocalization with an early endosome marker, the Early Endosome Antigen 1 (EEA1), and with a lysosome-specific marker, LAMP-1 (Figure 8). Indeed, there was colocalization of the Lamp-1 signal with both viral proteins F and N in A3.01 cells (Figure 8E–L), suggesting an involvement of lysosomes in protein trafficking or degradation. However, the imaging evidence is not enough to confirm specific involvement of lysosomal machinery in HRSV protein degradation, since the colocalizations of HRSV proteins with EEA1, an early endosome marker, was also found (Figure 8E,G,H,I,K,L). It is noteworthy that the same pattern was not observed in the colocalization of Lamp-1 and EEA1 as in the HRSV inclusion bodies in HEp-2 cells (Appendix A). To further investigate the association of the HRSV F and N proteins with cellular endosomal system, we also did an immunofluorescence microscopy for CD63 and Sorting Nexin 2 (SNX2). CD63 is a marker for late endosomes, while SNX2 is part of the cellular retromer complex responsible for carrying cargoes from early endosomes to the trans-Golgi [38,39]. In addition, it was recently found that SNX2 is recruited to HRSV N inclusion bodies and plays a role in the HRSV viral production in HEp-2 cells [24]. In A3.01 cells, HRSV F and N proteins form cytoplasmic granules within a large compartment occupying most of the cytoplasmic space. Several of these granules also contain CD63 (Figure 9E–N). SNX2 is also present in the same compartment, but apparently not in the same granules (Figure 9E–N). This is based on the colocalization analysis, which showed that that CD63 colocalized significantly more than SNX2 with HRSV proteins in A3.01 cells (Figure 9I,N). Therefore, in contrast to HEp-2 cells, the HRSV N-positive structures failed to recruit SNX2 in A3.01 cells (Figure 9O–Z).

### 3.9. The Production of HRSV Filaments at the Plasma Membrane of A3.01 Cells Is Very Low

HRSV assembly/budding takes place at the plasma membrane, with the appearance of viral protein-containing filament-shaped structures in infected epithelial cells [34,40].

Since the filaments emerging from the plasma membrane are one of the hallmarks of the HRSV infection [1], we examined whether HRSV is capable of producing typical filamentous structures in A3.01 cells. Immunofluorescence for viral proteins was done with special attention to the plasma membrane. Although only scarcely, HRSV N and M proteins could be seen in filament-shaped structures (Figure 10A–C, arrowheads). Nevertheless, the quantity of filament structures emerging from A3.01 cells was minimal when compared to those found in HEp-2 cells (Figure 10D–I). It is noteworthy that not only the filament formation was reduced in A3.01 cells (Figure 10A–C and G–I), but the quantity of the A3.01 cells displaying at least one filament for HRSV proteins was also significantly lower than that of HEp-2-infected cells (Figure 10G). Whereas 100% of infected Hep-2 cells had at least 30 filaments in all microscopic fields examined (Figure 10H), the percentage of infected A3.01 cells with filaments ranged from 0% to 100% depending on the microscopic field examined, with mean equal to 45% (Figure 10G), and all these cells harbored less than 10 filaments por cell (Figure 10H). Therefore, while HRSV proteins were readily detected in intracellular compartments of A3.01 cells, very little accumulation of HRSV products was seen at the plasma membrane, which resulted in a lower filament production in these cells.

## 4. Discussion

Studies on the interaction of HRSV with host cells are usually conducted using respiratory epithelial cells, which are the main targets of the natural infection that usually results in cell death [1]. However, HRSV is also able to infect nonrespiratory cells, including CD4^+^ T lymphocytes [2,3,5], and may cause persistence in some cell types, such as murine macrophages [41]. The frequent detection of HRSV RNA in tonsillar tissues from children without symptoms of acute HRSV infection [4] suggests that the agent may cause prolonged infection in secondary lymphoid tissues. It is thus presumable that HRSV causes patterns of infection that differ between lymphoid and epithelial cells, which are likely to affect functions and survival of these cells. The present study was undertaken to elucidate details of in vitro HRSV infection of the human CD4^+^ T cell line A3.01 in comparison to HRSV infection in the commonly used cell line HEp-2. We showed that HRSV infection in A3.01 cells is different from that in HEp-2 cells at multiple steps of the virus replication cycle.

Whereas HRSV was able to infect A3.01 cells, the progeny production in these cells was much lower than that of HEp-2 cells. This could be due to reduced susceptibility and/or permissiveness of A3.01 cells to infection by HRSV. In that regard, we showed that HRSV fusion with A3.01 cells was considerably reduced as compared to the HEp-2 cells. This could be due to some limitation of the HRSV fusion process itself or due to a difference in HRSV-receptor engagement in A3.01 cells, creating subsequent hindrance to the fusion process. It is noteworthy that HRSV can use different receptors to fuse to host cell membranes [10,42,43]. Therefore, while the binding of HRSV to HEp-2 and A3.01 cells are comparable, differences in the levels of expression or affinity of individual receptors between the two cell types may affect the efficiency of virus-cell fusion. It is also worthwhile to test the activity of macropinocytosis in A3.01 cells versus HEp-2 cells, since this is the best known mechanism of HRSV internalization [9,44].

The replication of the HRSV genome occurs in IBs [1]. Recently, Rincheval et al. demonstrated that the HRSV mRNAs are sequestered within the IBs, more specifically in organized structures that they called inclusion bodies-associated granules (IBAGs) [14]. Using IF and FISH approaches in the present study, we found that HRSV IBs in A3.01 cells are significantly less abundant, smaller, and morphologically different than those seen in HEp-2 cells, and that most of the IBs that are formed in A3.01 cells lack IBAGs. It is thought that in IBAGs, the M2-1 protein of the HRSV binds to viral mRNAs, thereby making them more stable, which consequently ensures a better protein production [14]. Therefore, it is likely that the lack of discrete HRSV IBAGs in A3.01 cells at least partially accounts for the reduction in HRSV proteins N, M, and P in A3.01 cells in comparison with HEp-2 cells at all times post-infection (Figure 4). HRSV proteins N and P are integral parts of the RNA transcription complex and IBs [45,46,47]. Therefore, lower levels of these proteins could help to explain the considerably reduced rates (about 10,000-fold) of HRSV genome production in A3.01 cells compared to the HEp-2 cells. Together, these findings lead us to speculate that the low permissiveness to the HRSV genome replication in A3.01 cells is a sum of defects in formation of IBs and IBAGs and protein translation, where inefficiencies in each one of these steps exacerbate the other.

Since HRSV uses the secretory pathway to deliver viral proteins to the assembly sites at the plasma membrane [1,15,17,19,23,24], we examined the presence of virus proteins along the main components of the secretory and endosomal pathways by immunofluorescence. In line with the previous literature, our results showed that HRSV F and N proteins partially colocalized with markers of the secretory pathway [1,23,24]. However, it is noteworthy that, in contrast to HEp-2 cells, A3.01 cells did not display an evident accumulation of SNX2 at places where the HRSV N protein was. In HEp-2 cells, SNX2 was found recruited to N-positive structures, and the knockdown of SNX2 negatively impacted the HRSV production [24]. The absence of the recruitment of SNX2 to N-positive structures in A3.01 cells could contribute to the inefficiency in the viral production in A3.01 cells. The colocalization of CD63 and Lamp-1 with HRSV proteins is consistent with viral protein degradation, which could explain the lower amounts of the HRSV proteins in A3.01 cells. However, more studies should be performed to specifically assess this question, since it is currently unknown whether the viral protein synthesis is impaired or whether they are degraded in this cell type.

Interestingly, even though the HRSV proteins partially colocalized with secretory pathway markers in A3.01 cells, the number of the A3.01-infected cells displaying filaments at its surface was dramatically low. Furthermore, the number of filaments per infected cell was at least 10-fold less in A3.01 than in HEp-2. These results suggest that the trafficking of viral components to the virus assembly sites is defective in A3.01 relative to HEp-2 cells. At this time, we do not rule out that the absence of viral filaments at the cell surface is due to a defect in the assembly process, which may be caused by low viral protein levels or by yet another block on the process at the plasma membrane.

In the present study, we did not evaluate HRSV infection in primary CD4^+^ T cells, what can be considered as a limitation. HRSV genome has been detected in secondary lymphoid tissues from children undergoing tonsillectomy [4], and HRSV infection of human CD4^+^ T cells has also been reported [5]. However, the biology of HRSV interaction with CD4^+^ T cells has not been investigated. In that regard, and despite the differences between A3.01 and normal primary CD4^+^ T cell cultures, the authors of [5] found that primary CD4^+^ T cells did not exhibit replication in the first 48 h of infection, which is similar to our results. The permissiveness of primary CD4^+^ T cells was also similar to our findings in A3.01 cells. To some extent, these similarities give strength to our study, suggesting that HRSV infection of A3.01 cell line can be a practical model to study details of HRSV infection of CD4^+^ T cells.

## 5. Conclusions

Overall, the present results show that HRSV infection of A3.01 human CD4^+^ T cell line is virtually unproductive, with only insignificant virus production as compared to HEp-2 cells. This is due to multiple defects during HRSV replication in A3.01 cells, namely low virus-cell fusion, formation of hypofunctional inclusion bodies lacking IBAGs, failure to achieve high viral protein levels, and possibly altered trafficking of viral proteins and genome to virus assembly sites at the plasma membrane.

## Figures and Tables

**Figure 1 viruses-13-00231-f001:**
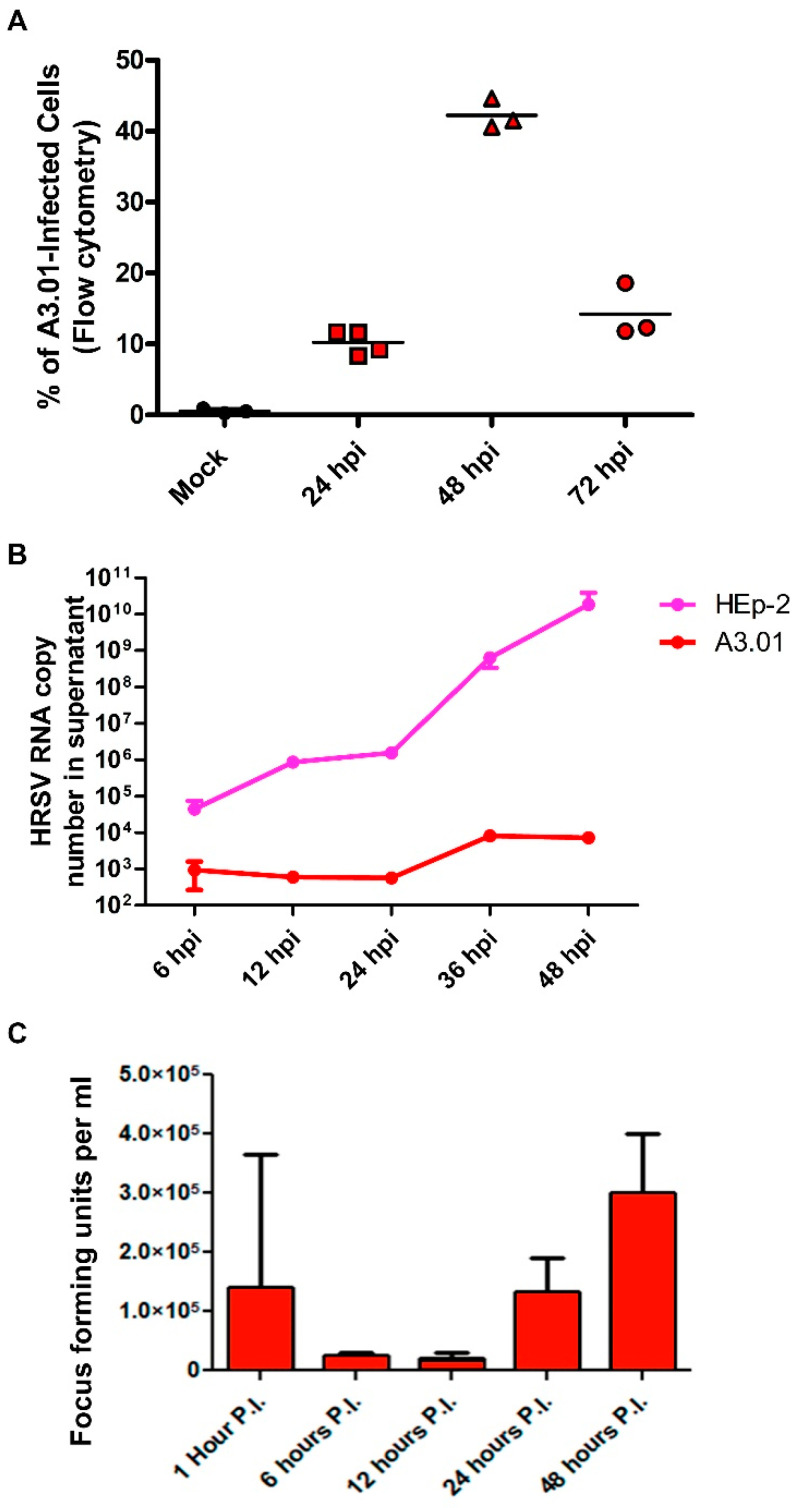
Infection of human respiratory syncytial virus (HRSV) in A3.01 cells. (**A**) Flow cytometry analysis of mock and HRSV-infected cells showing the percentage of the infected cells by detection of HRSV N protein. (**B**) RT-qPCR of HRSV genome in supernatant of infected HEp-2 and A3.01 cells over time post-infection. (**C**) HRSV progeny production in A3.01 cells determined by fluorescent focus assay. All results are from at least three independent experiments.

**Figure 2 viruses-13-00231-f002:**
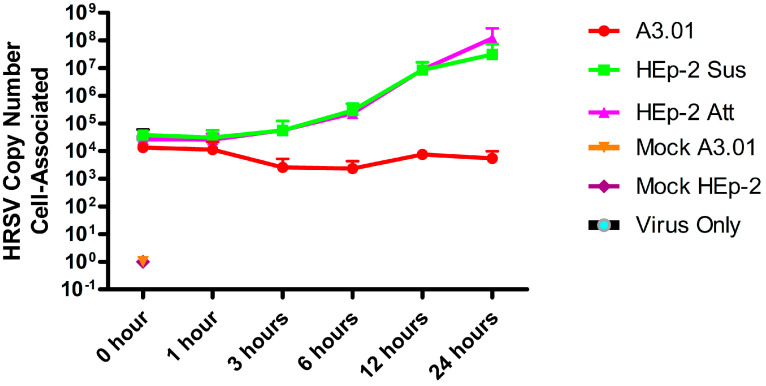
Intracellular accumulation of HRSV genome in A3.01 and HEp-2 cells. A3.01 and HEp-2 cells attached (Att) or in suspension (Sus) were inoculated with HRSV or mock-inoculated and kept at 4 °C for 1 h for attachment. Then, cells were centrifuged and collected for qPCR analysis at time zero and at different times thereafter. Genome quantification was plotted in the Y axis. The “virus-only” well received only virus in the absence of cells. This graph is a representation of three independent experiments.

**Figure 3 viruses-13-00231-f003:**
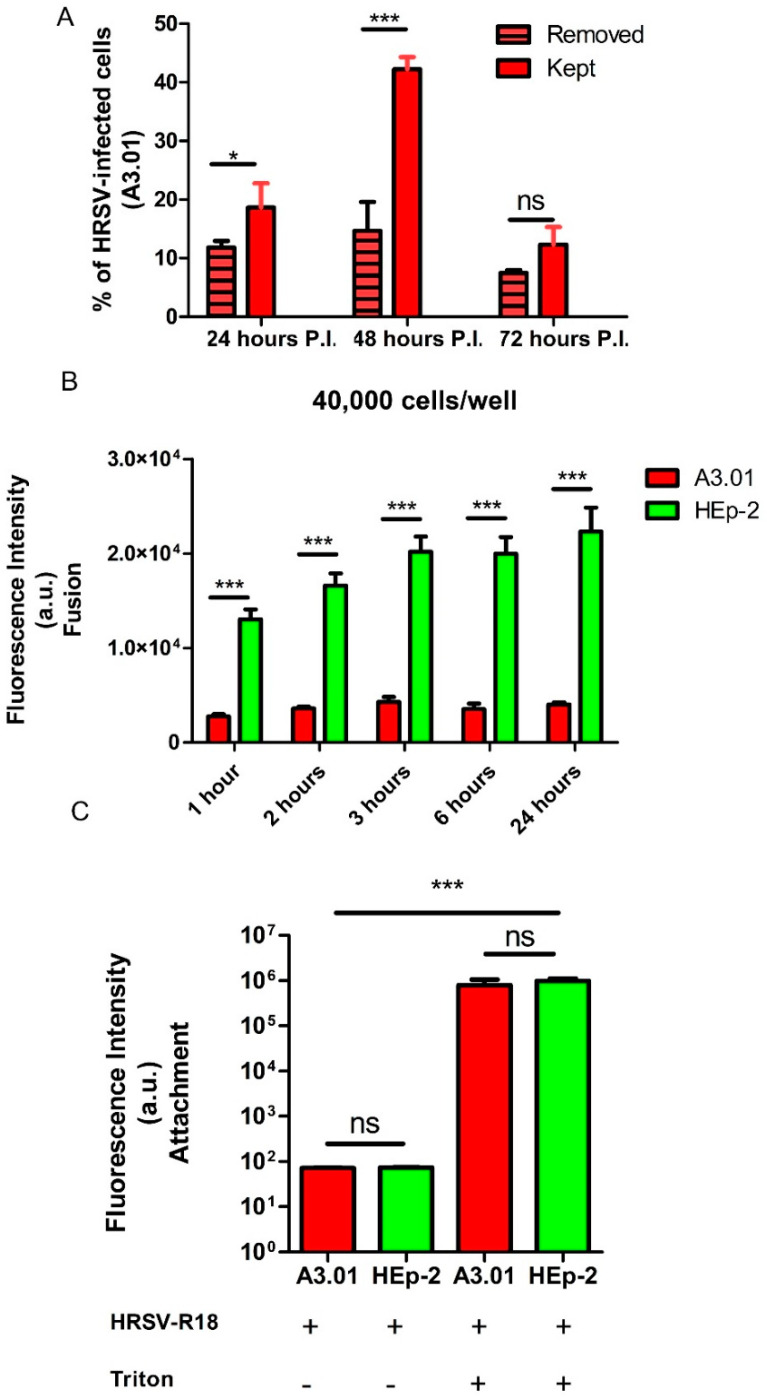
The fusion process in A3.01 is inefficient. (**A**) Differences between HRSV-infected cells with and without inoculum removal at different times post-infection. (**B**) Comparison of the fusion process in A3.01 and HEp-2 cells over time with 40,000 cells per well. (**C**) Comparison of the HRSV attachment in A3.01 and HEp-2 cells. The graphs in (**A**), (**B**), and (**C**) represent at least three independent experiments. The statistical method used was student’s *t*-test, * *p* < 0.05 and *** *p* < 0.001. The intensity of fluorescence emitted by R18 was measured by a SynergyTM Multi-Mode Microplate Reader.

**Figure 4 viruses-13-00231-f004:**
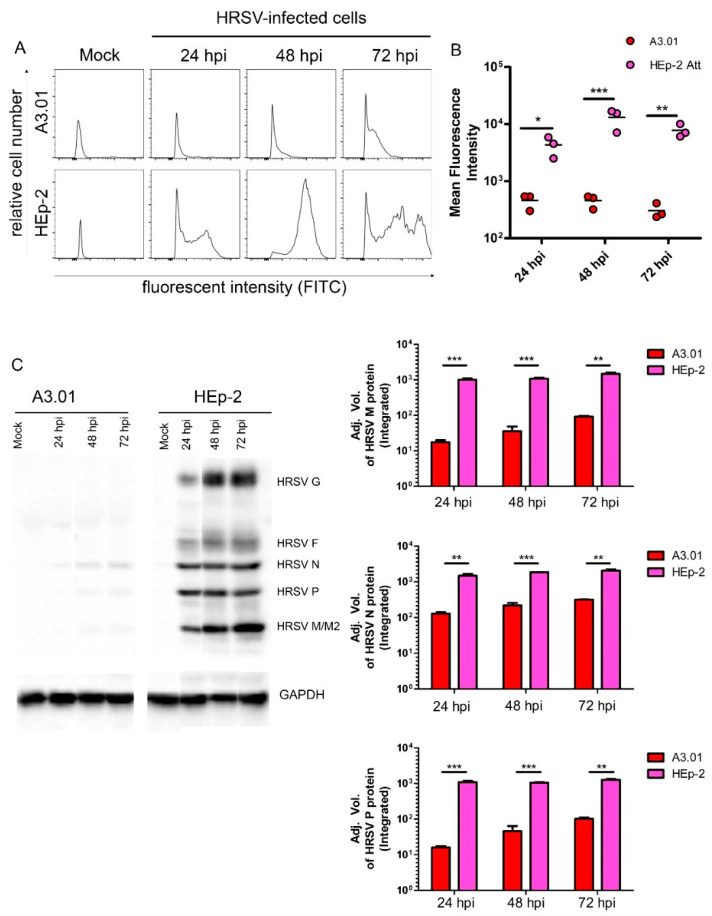
HRSV protein production in A3.01 cells is discrete. (**A**) Histogram of mean intensities of fluorescence of cells by a flow cytometry. (**B**) Graph plotted from three different experiments comparing the production of HRSV N protein in A3.01 and HEp-2 attached (Att) cells at 24 h, 48 h, and 72 h post-infection. (**C**) Western blot of HEp-2 and A3.01 cells infected or not (MOCK) by HRSV. GAPDH was used as housekeeping loading control, and the graphs represent the analysis of the protein bands of three independent experiments. The statistical method used in (**B**) was two-way ANOVA, and the statistical method used in (**C**) was student’s *t*-test, * *p* < 0.05, ** *p* < 0.01, and *** *p* < 0.001.

**Figure 5 viruses-13-00231-f005:**
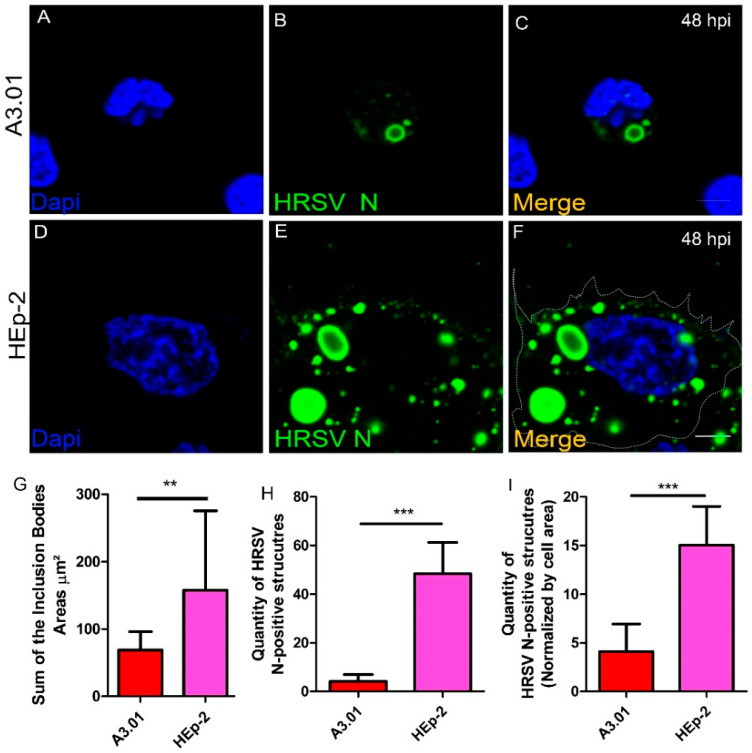
HRSV inclusion bodies in HRSV-infected A3.01 and HEp-2 cells. (**A**–**C**) A3.01 cells stained for HRSV N protein at 48 hpi (green fluorescence). (**D**–**F**) staining for HRSV N protein in HEp-2 cells at 48 hpi. (**G**) Comparative analysis of the sum of the inclusion body area in A3.01 and HEp-2 cells. (**H**) Comparative analysis of numbers of vesicular structures stained for HRSV N in A3.01 and HEp-2 cells. (**I**) Comparative analysis of the structures stained for HRSV N in A3.01 and HEp-2 cells normalized by cell area. The immunofluorescence images shown in figures (**A**–**F**) represent a single focal plane of at least three independent experiments. The images were taken in a Zeiss 780 confocal microscope. Magnification 63x. The scale bars represent = 10µm. The graphs in (**G**) and (**H**) represent at least three independent experiments. The statistical method used was student’s *t*-test, ** *p* < 0.01 and *** *p* < 0.001.

**Figure 6 viruses-13-00231-f006:**
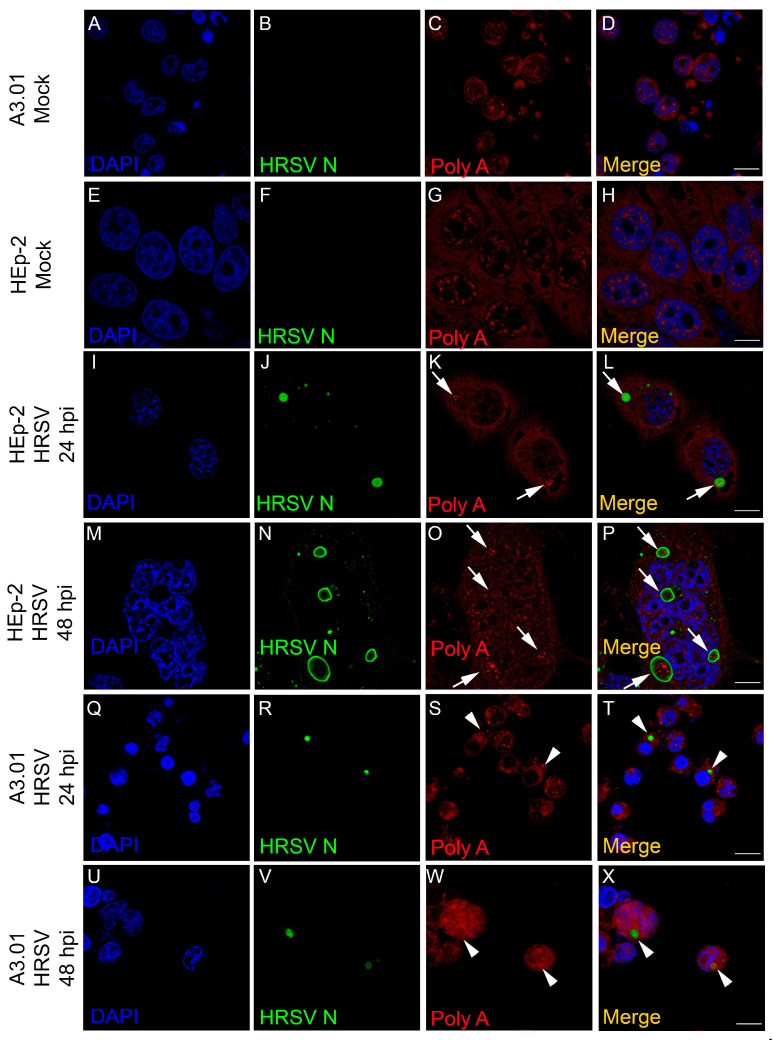
HRSV inclusion bodies in A3.01 in the absence of IBAGs. (**A**–**D**) and (**E**–**H**) A3.01 and HEp-2 mock-infected cells. (**I**–**P**) HRSV infected HEp-2 cells at 24 and 48 hpi, as shown by inclusion bodies (IBs) in (**J**,**N**), respectively. Within the IBs in (**J**,**N**), it is possible to see a fluorescent in situ hybridization (FISH) signal for the IBAGs, as shown in (**K**,**L**,**O**,**P**) (arrows). (**Q**–**X**) represent A3.01-infected cells at 24 and 48 hpi. No inclusion body-associated granules (IBAGs) were seen within HRSV inclusion bodies (**R**,**V**) in these cells, as shown in (**S**,**T**,**W**,**X**). This set of figures represents a single focal plane of three independent experiments taken in a Zeiss 780 Confocal. Magnification 63x. Scale bars = 10 µm.

**Figure 7 viruses-13-00231-f007:**
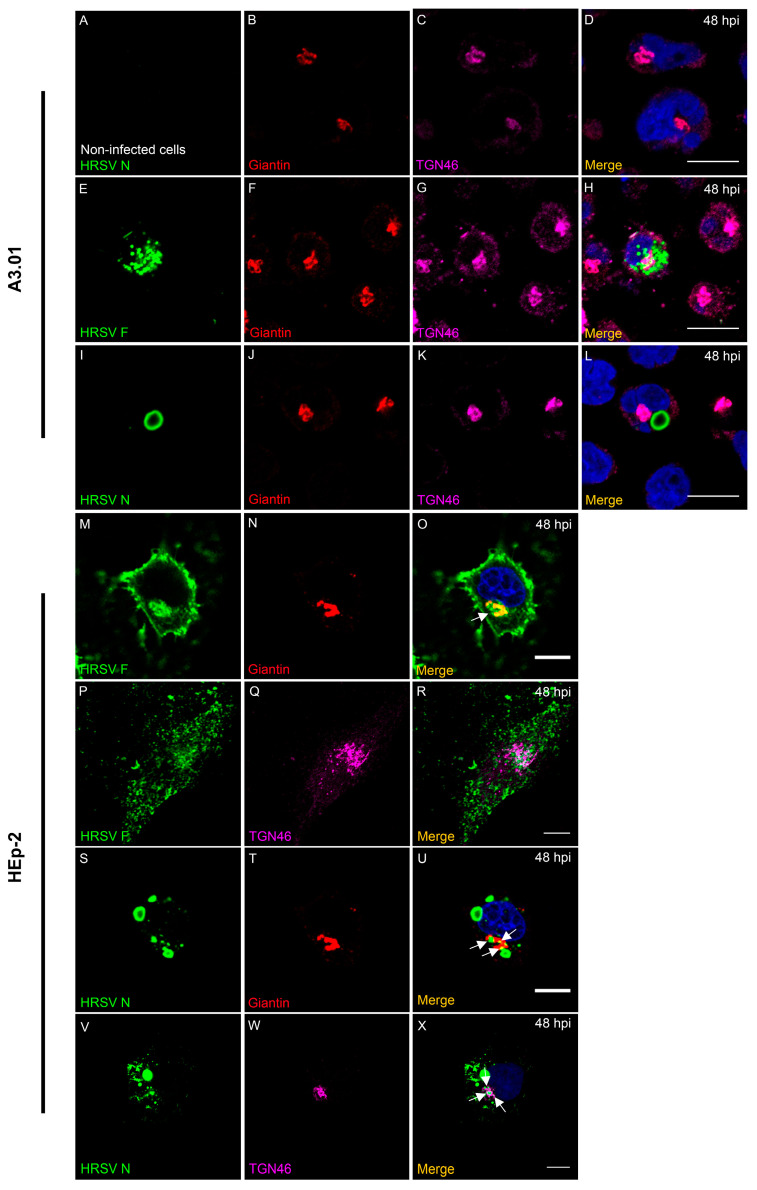
Colocalization analysis of HRSV proteins at the Golgi in A3.01 and HEp-2 cells. (**A**–**D**) A3.01 mock-infected cells stained for cis and medial-Golgi (Giantin) in red (**B**), trans-Golgi (TGN46) (magenta) (**C**), merge (**D**). (**E**–**H**) HRSV-infected A3.01 cells at 48 hpi stained for HRSV F (green) (**E**), giantin (**F**), and TGN46 (**G**). The merge is depicted in (**H**). (**I**–**L**) HRSV-infected A3.01 cells at 48 hpi stained for HRSV N (**I**), giantin (**J**), and TGN46 (**K**). The merge of the figure is in (**L**). (**M**–**O**) HRSV-infected HEp-2 cells at 48 hpi stained for HRSV F (green) (**M**), giantin (red) (**N**), and merge (**O**). The arrow points to co-localization. (**P**–**R**) HRSV-infected HEp-2 cells at 48 hpi stained for HRSV F (**P**), TGN46 (magenta) (**Q**), and merge (**R**). The arrow points to colocalization. (**S**–**U**) HRSV-infected HEp-2 cells at 48 hpi stained for HRSV N (**S**), giantin (**T**), and merge (**U**). The arrows point to colocalization. (**V**–**X**) HRSV-infected HEp-2 cells at 48 hpi stained for HRSV N (**V**), TGN46 (**X**), and the merge (**X**). The arrows point to colocalization. All the figures represent a single focal plane of at least three independent experiments and Z-stack taken in a Zeiss 780 Confocal or Leica Sp5 Confocal microscope. Magnification 63×. Scale bars = 10 µm.

**Figure 8 viruses-13-00231-f008:**
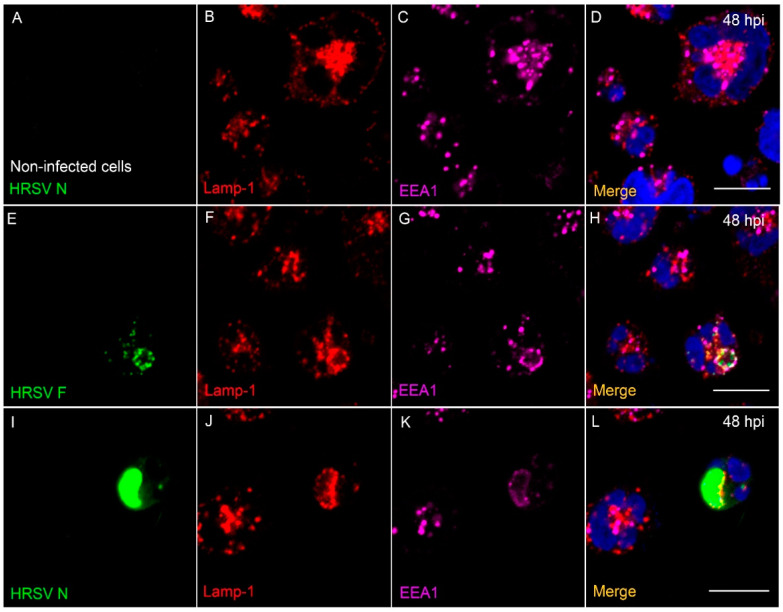
Colocalization of EEA1 and Lamp-1 with HRSV proteins in A3.01 cells. (**A**–**D**) A3.01 mock-infected cells. (**B**) Lamp-1 (red), (**C**) EEA1 (magenta), and (**D**) the merge of the set of the figures. (**E**–**H**) HRSV-infected A3.01 cells at 48 hpi, stained by HRSV F (green) (**E**), Lamp-1 (**F**), and EEA1 (**G**). The merge to this set of figures is depicted in (**H**). (**I**–**L**) HRSV-infected A3.01 cells at 48 hpi, stained by HRSV N (green) (**I**), Lamp-1 (**J**), and EEA1 (**K**). The merge of this figure set is depicted in (**L**). All the images were taken in a Zeiss 780 Confocal and are a representation of a single focal plane. Magnification 63×. Scale bars = 10 µm.

**Figure 9 viruses-13-00231-f009:**
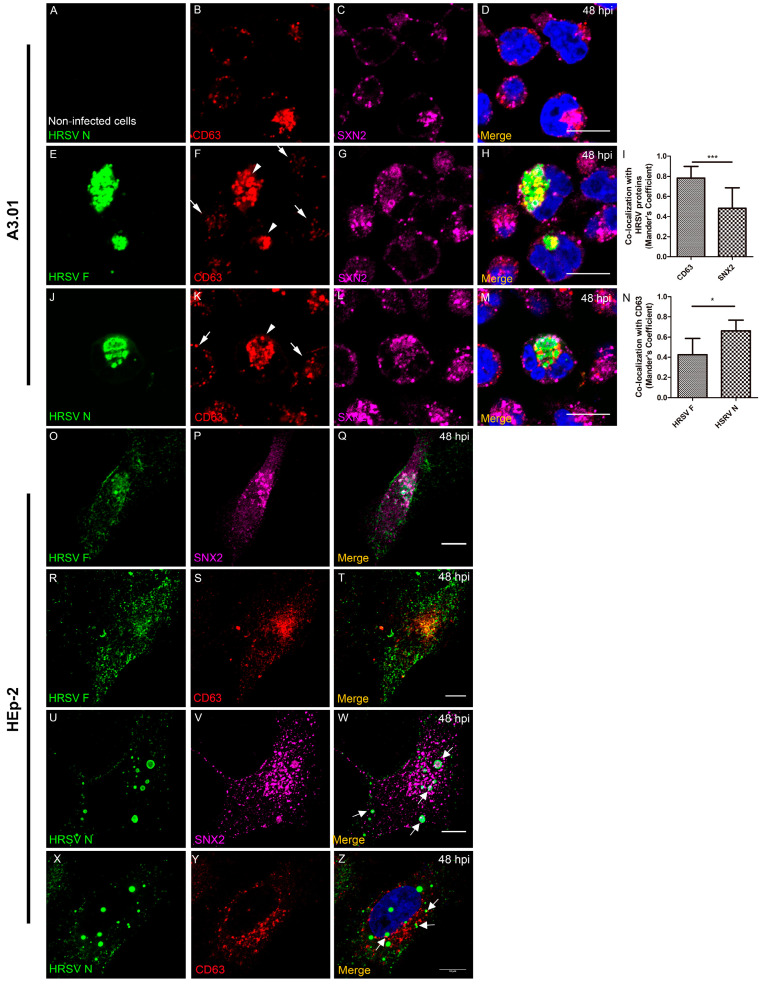
Colocalization of CD63 and SNX2 with HRSV proteins in A3.01 and HEp-2 cells. (**A**–**D**) A3.01 mock-infected cells at 48 hpi. (**B**) CD63 (red), (**C**) SNX2 (magenta), and (**D**) the merge. (**E**–**H**) HRSV-infected A3.01 cells at 48 hpi, stained by HRSV F (green) (**E**), CD63 (**F**), and SNX2 (**G**). In (**F**), the arrowheads point to the places where the HRSV F protein was found and the CD63 was accumulated. The arrows point to the places where the cells were not infected and there was not CD63 accumulation. The merge to this set of figures is depicted in (**H**). (**I**) Graph of colocalization between CD63 and SNX2 with HRSV proteins in A3.01 cells, showing significant colocalization of CD63. (**J**–**M**) HRSV-infected, A3.01-infected cells at 48 hpi, stained by HRSV N (green) (**J**), CD63 (**K**) and SNX2 (**L**). In (**K**), the arrowhead points to the place where the HRSV N protein was found and the CD63 was accumulated. The arrows point to the places where the cells were not infected and there was not CD63 accumulation. The merge to this set of figures is depicted in (**M**). (**N**) Graph comparing the colocalization of HRSV F and N proteins with CD63. (**O**–**Q**) HRSV-infected HEp-2 cells at 48 hpi, stained by HRSV F (green) (**O**), SNX2 (magenta) (**P**), and the merge (**Q**). (**R**–**T**) HRSV-infected HEp-2 cells, stained by HRSV F (**R**), CD63 (red) (**S**), and the merge (**T**). (**U**–**W**) HRSV-infected HEp-2 cells at 48 hpi, stained by HRSV N (green) (**U**), SNX2 (**V**) and the merge (**W**), where the arrows point to colocalization. (**X**–**Z**) HRSV-infected HEp-2 cells at 48 hpi, stained by HRSV N (**X**), CD63 (**Y**), and the merge (**Z**). All the images were taken in a Zeiss 780 Confocal or Leica SP5 Confocal and are a representation of a single focal plane of Z-stack or not experiments. Magnification 63×. Scale bars = 10 µm. The graphs shown in figures I and N represent the Mander’s Coefficient analysis based on three or more independent experiments and were done in at least five cells per field. The statistical method used was student’s t-test, * *p* < 0.05 and *** *p* < 0.001.

**Figure 10 viruses-13-00231-f010:**
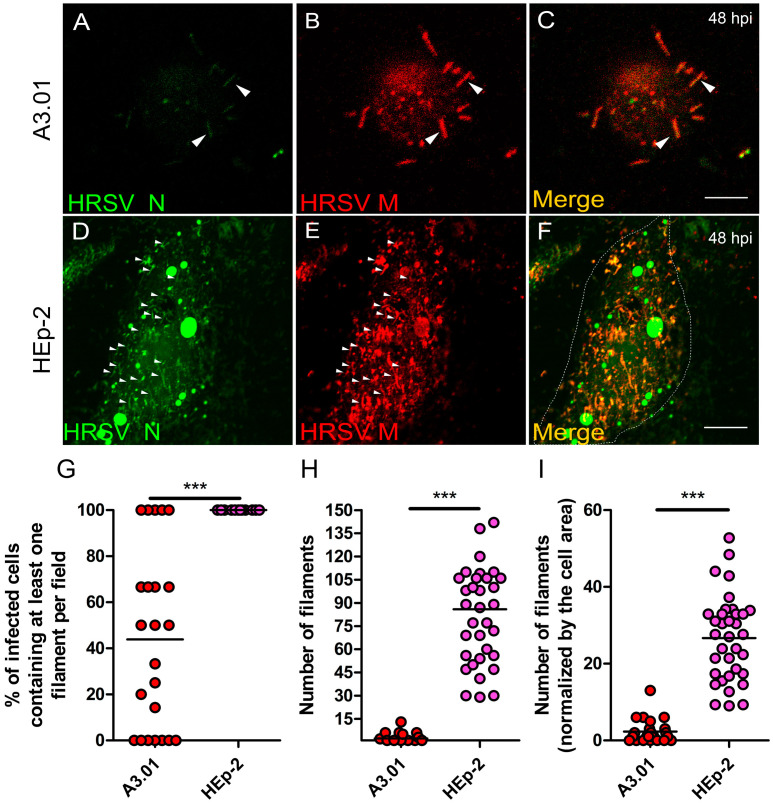
Filament formation of HRSV in A3.01 is rare. (**A**–**C**) Immunofluorescence for HRSV N and M in A3.01-infected cells at 48 hpi, depicting some filaments pointed out by arrowheads. (**D**–**F**) Immunofluorescence for HRSV N and M in HEp-2-infected cells at 48 hpi, depicting filaments pointed out by arrowheads. (**A**–**F**) A single focal plane of at least three independent experiments taken in a Leica SP5 Confocal. Magnification 63×. This experiment was repeated at least three independent times. The scale bar of figure (**C**) = 10 µm. (**G**) Graph of the percentage of A3.01 and HEp-2-infected cells displaying at least one filament emerging from the plasma membrane per field. (**H**) Graph of the quantity of the filaments in A3.01 and HEp-2-infected cells. (**I**) Graph of the quantity of the filaments in A3.01 and HEp-2-infected cells normalized by the cell area. The graphs depicted in (**G**), (**H**), and (**I**) are representative of more than five independent experiments. Each dot in the graphs (**G**), (**H**), and (**I**) corresponds to one microscopic field. The statistical method used was student’s *t*-test, *** *p* < 0.001. All the images were taken in a Zeiss 780 or Leica SP5 Confocal and are a representation of a single focal plane. Magnification 63×. Scale bars = 10 µm.

## Data Availability

All relevant data are contained within the article or Appendix A. Any additional data are available on request from the corresponding author.

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
