# Peer review of "Human Respiratory Syncytial Virus Infection in a Human T Cell Line Is Hampered at Multiple Steps"

_viruses, 2021, doi:10.3390/v13020231_

Round 1
Reviewer 1 Report
In the submitted manuscript “Human respiratory syncytial virus infection in a T cell line is hampered at multiple steps”, Cardoso et al. investigated the replication of HRSV in A3.01 cells compared to the HEp-2 cells. They used different tools such as RT-qPCR, flow cytometry, immunofluorescence, and co-localization to show that A3.01 cells do not support robust HRSV propagation due to factors including low viral fusion and hypo-functionality of inclusion bodies. The study is interesting, but some comments still need to be fulfilled.
Comments:
- The title: please change the term “a T cell line” to “A3.01 human T cells”.
- Line 15: add the (HRSV) immediately after the full name Human respiratory syncytial virus.
- Line 24: it is RT-qPCR and not qPCR.
- Line 113: define PFA.
- Line 115: which antibody dilution and which diluent were used, please clarify.
- Line 116: define BSA.
- Line 128: is it in PBB 1X or PBS, please check?
- Line 131-144, please add the working dilutions of all antibodies.
- Line 131-144, how the authors avoided the background, was there a negative control for staining?
- Line 139: put the source of ImageJ software.
- Line 145, add all antibody dilutions used in Fluorescent in situ hybridization.
- Line 161: change real time PCR to “real time RT-PCR”.
- Line 161-171: all information regarding the reaction mixtures of the RT and PCR should be included. For example, the amount of RNA, the primer sequences, the probe used, the name of the kits, enzymes, the real time machine, and the limit of detection of the assay.
- Line 174: change “Evelyn M. Covés-Datson” to “Covés-Datson” et al.
- Line 185-193, please mention in this paragraph if there was any washing after infection or just incubated for 72 hours without washing.
- Line 185-193, the dilutions of antibodies should be mentioned.
- Line 190: change of 100X Triton to 0,01% triton X-100.
- In the materials and methods section, the western blotting is missing. This should be included with a clear description for the amount of proteins or cell lysates used, antibody dilutions, incubations periods, visualization process, etc.
- In the materials and methods section, the statistical analyses part is missing. Please clearly describe how the statistical analyses were performed.
- In figure 1B, how many replicates were used for quantification, the error bars are not visible at all time points.
- Line 212: RT-qPCR instead of qCR
- Line 224: either use “quantitative RT-PCR” or “real time RT-PCR”
- Line 272: it is mentioned “40000 cells” while in line 256, mentioned as 4x103. Please check and clarify?
- Figure 4C: the left panel is very faint and bands are not clearly visible. Why the background of the parts are different. Better figure is needed. Also, bands of the house keeping gene should be included in both cell lines.
- Use either IBs or IB's, and fix it in the whole manuscript.
- The authors need to read the manuscript and check for typos.
Author Response
The attached file contains the answer for referee's 1 questions, the updated version of the full manuscript after the modifications and the updated version of the manuscript figures.

Reviewer 2 Report
In this manuscript by Ricardo de Souza Cardoso et al., the authors examined the replication of human Respiratory Syncitial Virus (HRSV) in lymphoid cells, and more specifically in the human T lymphocyte cell line A3.01. The authors focused on the replication steps and assembly of HRSV in A3.01 cells.
Main findings
- HRSV-infected A3.01 cells are inefficient in virus production (as demonstrated by (i) the low rate of antigen positive-cells, (ii) the very low rate of viral genome multiplication and (iii) the very low yield of infectious virus in the supernatant).
- While the virus adsorbs to A3.01 cells, there seems to be a defect in its fusion process- The synthesis of viral proteins in the infected cells is very low, when compared with the similarly infected Hep-2 cells.
- The formation of inclusion bodies is compromised in A3.01, and these inclusion bodies lack the structuration that is observed in similarly infected Hep-2 cells.
- While the viral F and N proteins colocalize with the Golgi and with markers of the endosomal pathway in Hep-2 cells, this colocalization is essentially absent in A3.01 cells.
- Finally, infected A3.01 produced much less viral filaments at their surface than did similarly infected Hep-2 cells.
- Therefore, the replication of HRSV in A3.01 cells is defective at several steps of the viral cycle (fusion, genome replication, formation of inclusion bodies, recruitment of cellular proteins, virus assembly and budding).
The manuscript is interesting, but there are several problems that need to be addressed before it can be considered for publication. Notably, while the observations seem clear, they are not accurately described in the text (the text is somewhat imprecise or does not faithfully/exactly reflects what is observed in the figures). Further, several interpretations rely on the microscopic observation of a single cell. The authors should try to represent larger microscopic fields, showing several cells.
This is not in conflict with the interpretation, but instead the interpretation would be clearer with more accurate observations. Below are the major remarks, and a few minor remarks.
Major remarks
Line 88 and 196… in the human cell line A3.01; ..A3.01 human lymphocytes…(the authors should clearly indicate that A3.01 cells are of human origin).
All graphs. Keep the same color code for all the graphs (Hep-2 in green or blue, A3.01 in red).
Figure 1B. Label for the y-scale should be “HRSV RNA copy number in supernatant”. And the y-scale should begin at 10^2 (in order to enlarge and to better visualize the increase between 24 and 36h).
Figure 1A and 1C. Since the data represent 3 independent experiments, the authors should replace the bar graph by a scatter dot plot (three dots per time point, possibly with a bar representing the median).
Figure 1B. Since the data represent 3 independent experiments, the authors should add error bars to each dot (and clearly indicate that the dots represent the means).
Figure 1C. The label of the y-scale should indicate “Focus forming units per ml” (i.e. per ml of cell supernatant).
Line 232. …but while the virus replicated about ten thousand folds in HEp-2 cells, we observed no replication in A3.01 cells (no increase in the amount of viral RNA)…
Lines 260-261…. while the quantity of de-quenched R18 steadily increased over time in HEp-2 cells, it remained at its background level in A3.01 cells.
Line 268…is nearly abolished in comparison to HEP-2 cells.
Figure 4B and line 303. The y-scale should be limited to [10^2-10^4 or 10^5], and the bar-graph should be replaced by a scatter dot plot (three dots per cell per timepoint, representing the three experiments). To emphasize the logarithmic nature of the y-scale, add the subdivisions (2 to 9)
Line 286-87… the N protein level in each HRSV-infected A3.01 cell remained close to the background level, and at least ten to a hundredfold lower than that in each HEp-2-infected
Lines 290-91…. revealed that only the N protein could be reliably detected in infected A3.01 cells at different times post-infection, while only faint bands revealed small amount of the other viral proteins
Figure 4C. The authors should choose a logarithmic y-scale (with subdivisions) for the three graphs, in order to more accurately quantify the low levels of the three proteins (which are not zero but seem to slightly increase over time)
Lines 310-322 and Figure 5. The text relies on the interpretation on only a single infected cell. The authors should show a larger field with several infected cells (perhaps with an inset showing a single cell).
Line 316 and Figure 5G…. measuring the areas of IBs in A3.01 (the authors should try to represent the areas instead of the perimeters)
Lines 353-354…. While we observed, on average, zero or one infected A3.01 cells containing IBs per microscopy field, the average was 15 IBAG-containing cells per field for Hep-2 cells (suppl. Figure 1D). Further, while each IB contained 3 to 4 IBAGs in Hep-2 cells, no authentic IBAG could be clearly identified in the rare IBs of the infected A3.01 cells.
Lines 375-386. Since this paragraph deals with the comparison of A3.01 and Hep-2 infected cells, it would be more convenient (in fact it is necessary) to compare the photographs between two adjacent or successive figures (instead of comparing figure 7 E-H with suppl. Figure 2A-F, then figure 7I-L with suppl. Figure 3A-F). Further, virtually all of these figures (Figures 7, 8, 9, 10, and suppl. Figures 2 and 3) show a single cell per panel, precluding any generally valid interpretation (see notably suppl. Figures 2ADGJM and 3ADGJM). For instance the subcellular distribution of F seems very different between suppl. Figure 2A and 2D. Also, as regards Figure 7 and suppl. Figures 2 and 3, the panels should be arranged similarly in order to allow comparisons (e.g. for instance one should be able to easily compare Figure 7EFGH with similarly arranged panels for infected Hep-2).
Lines 377-79. Contrary to the interpretation of the authors, it seems rather that F is present in dots that are distributed in the whole cytoplasmic volume (Fig. 7E-H), with no obvious association with the Golgi, even partial.
Lines 380-81. In Hep-2, the cytoplasmic fraction of F (as opposed to the large fraction that is membrane-associated [suppl. Fig 2A]) shows strong colocalization with giantin. However, contrary to suppl. Figure 2A, there is no such membrane-associated fraction of F in suppl. Figure 2D.
Lines 382-83. There is no convincing co-localization of N and TGN46 in Hep-2 cells (suppl. Figure 3DEF).
Lines 384-86. The whole final sentence of this paragraph should be reviewed, in order to more faithfully interpret what is observed and what distinguishes A3.01 and Hep-2 as regards the subcellular localization of F and N.
Lines 397-413 and Figure 8. Same remark as above, one cannot compare A3.01 and Hep-2 cells as regards the colocalization of F and N with EEA1 and Lamp1 (and there is no EEA1 panel for Hep-2 in suppl. Figures 2 and 3).
Figure 8MN and lines 416-420. There is no legend for the figures 8M, 8N. Further, these panels are of little use, and it is not known whether the co-localization coefficient rely on images from a single cell or from several infected cells.
Lines 427-430. Based on the observation of Figure 9, I would rather say that in A3.01 cells F and N form cytoplasmic granules within a large compartment occupying most of the cytoplasmic space. Several of these granules also contain CD63. SNX2 is also present in the same compartment, but apparently not in the same granules [this is in agreement with the sentence, lines 433-34]. However, this granular distribution of N in figure 9I contrasts with the structures shown in figures 8I and 7I in similarly infected A3.01 cells. Why? Again this emphasize the need to show several infected cells.
The picture is different in Hep-2 cells: F and N seem to co-localize partially with SNX2 (suppl. Figures 2GHI and 3GHI) but not with CD63 (suppl. Figures 2JKL and 3 JKL). Here again, a reliable comparison would require that similarly organized figures be put side by side for A3.01 and Hep-2.
Lines 438 and 441. What is the meaning of “The arrows point to the places where the cells are not infected”? Line 464. “..is significantly lower than that….” gives no useful information. It would be much more informative to say “While 100% of the infected Hep-2 cells had at least 30 filaments, in all of the microscopic fields examined, the percentage of infected A3.01 cells with filaments ranged from zero to 100, depending of the microscopic field examined (mean =45%)”. But what do the dots represent in Figure 10H? Does one dot correspond to one microscopic field? And finally it would be much more informative to say (based on Figure 10I) “On average, each infected Hep-2 exhibited 80 filaments [range 30-140], while each A3.01 cell exhibited only eight filaments”.
Line 540. ….at its surface was dramatically low, and furthermore the number of filaments per infected cell was at least tenfold less in A3.01 than in Hep-2.
Line 547….unproductive, with only unsignificant virus production as compared to Hep-2.
Discussion. The authors should discuss the possible limitations of their study, notably in relation with references 2,3,5 that show HRSV infection of T cells. Could there be a specific defect in A3.01 cells relative to human CD4 cells? Or, alternatively, is it possible that in vitro infection of A3.01 cells more faithfully reflects the natural infection of T cells in vivo (as opposed to in vitro infection of Hep-2 which would allow an exceptional and unrealistic rate of replication)?
Minor remarks
Line 227. ..incubated for viral adsorption…
Line 407… of Lamp-1 signal with both viral proteins F and N in A3.01 cells…
Figure 8. The Lamp-1 and EEA1 labels are difficult to read in the corresponding panels.
Figures 10 HIJ. The error bar is unnecessary. What is the middle bar? Mean or median? Figure 10I. Number of filaments per infected cell (y label). Figure 10J, y-scale. What is the meaning of “number of filaments normalized by the cell size”? And what is the unit of the scale?
Line 518 “….could help to explain significantly reduced rates…”. Again, this gives little useful information. It would be much more informative to say “…could help to explain the considerably reduced rates (about ten-thousandfold) of genome production…” (or else “virtually abolished genome production”)
Author Response

(The authors gave the same response as above.)

Round 2
Reviewer 1 Report
I would like to thank the authors for addressing all the comments
Author Response
Answer: We want to thank the reviewer for important observations that helped us to improve the impact of the manuscript and make it clearer.
Reviewer 2 Report
(review of the revised manuscript). I thank the authors for having addressed all the points that I raised. However, I have still some remarks that need to be addressed and that should help the authors improve their manuscript.
First, the authors should have used the Viruses template (like they did for the first version).
Major remarks
Page 2, line 23. the fusion process of HRSV in A3.01 cells is nearly abolished in comparison
Page 2 line 24 the replication of HRSV in A3.01 cells was considerably reduced
Page 2 lines 31-34 I think it would be more informative to write “HRSV infection of A3.01 CD4+ T cells is virtually unproductive as compared to HEp-2 cells, as a result of defects at several steps of the viral cycle: fusion, genome replication, formation of inclusion bodies, recruitment of cellular proteins, virus assembly and budding.”
Figure 1A. With the scatter dotplot, the mean bar is sufficient. Since all the points are displayed, the error bar is unnecessary (it may even be annoying).
Figure 3C (and also Fig. 4 BC). Do not truncate the y-scale (to my opinion, truncated scales should be forbidden). And remove all the “1.0x” on the labels of the y-scale (10^0, 10^1, … 10^7 is much easier to read, like in Fig. 1B and Fig. 2)
Figure 4B. Do not truncate the y-scale; remove the error bars (with all the data points displayed, the error bars are useless).
Figure 4C (graphs). (i) do not truncate the y-scale; (ii) remove all the “1.0x” (i.e. simple scale 10^0 to 10^3 or 10^4) and (iii) add the subdivisions (2 to 9) between the powers of ten, to emphasize the log scale (subdivisions like those seen in the x-scale of Fig. 4A).
Page 17 line 325….but also other viral proteins were about ten times less abundant in A3.01 than in Hep-2 at 48 and 72h p.i. (hence the need of a non-truncated log scale to better visualize that difference, which is even ~one hundred fold at earlier times p.i.)
Lines 346-47….we found that they were much smaller than those in HEp-2 cells (Fig. 5G, and compare Fig. 5 BC to 5 EF) and about tenfold less abundant in A3.01 than in HEp-2 cells (figure 5H)
Lines 386-88….Further, the quantity of the IBAGs found in inclusion bodies was significantly lower in A3.01 cells than in HEp-2 cells (supplementary figure 1D). This sentence seems to repeat the one just before.
Lines 408-411. It would perhaps be better to reverse the order of the two sentences (first, Hep-2, second, A3.01), but this would also impose to change the order of the figures. (For instance: “While in HRSV-infected HEp-2 cells, the viral protein F clearly co-localized with giantin and partially with TGN46, in A3.01 cells we observed no obvious association of F with these Golgi markers at 48 hours post-infection (Figure 7E-H).”
Lines 467-68 (legend to figure 8). If I understand, this figure shows only A3.01 cells, contrary to what the title says. I think it would be better if the reader can compare with Hep-2. Therefore, even in spite of the absence of EEA1 labelling, the authors could keep (in the supplementary data) the co-localization of N with Lamp-1 in HEp-2 cells (and accordingly add a short comment in the main text, lines 448-49).
Line 519… examined, with mean equal to 45% (figure 10H), and all these cells harbored less than ten filaments per cell (Fig. 10I)
Line 557… fusion with A3.01 cells was considerably reduced as compared
Minor remarks
Page 5 line 98 … Rabbit polyclonal antibodies (remove space)
Figure 1A and line 196 on the same page: there are 4 data points at 24 hours.
Page 14 line 264. …incubated for viral adsorption for 1 hour
Page 15 line 286 … unbound viruses
Lines 451-52. The sentence is grammatically incorrect “since the co-localizations of HRSV proteins with Lamp-1 and EEA1, an early endosome marker”
Line 461… This is based on the co-localization
Line 514… in A3.01 cells (figures 10A-C and H-I), also the quantity
Figure 10 HIJ (and suppl. Figure 1 CD). Error bars are useless.
Line 610.. undergoing tonsillectomy (add a space).
Line 614. There is no reference to support that statement. Is it ref 5?
Finally, please carefully read again the manuscript (including the English grammar), I may have missed some minor points.
